# A sequential three-way decision model based on linguistic Z-numbers

Yi Mao[1,3,4]*, Yaning Xu[2,3,4], Yuezhong Fan[3,4]

**1** School of Economics and Management, Xi'an Shiyou University, Xi'an, China, **2** School of Economics and Management, Xi'an Technological University, Xi'an, China, **3** Ni Guangnan Academician Workstation, Xi'an Silk Road Internet of Things Industrial ParkManagement Co., Ltd, Xi'an, China, **4** Base of Digital Applied Science Popularizing, Researching and Talent Cultivating in Chan-BaEcological District of Xi'an Silk Road Internet of things Industrial Park Management Co., Ltd, Xi'an, China

* maoyi@xsyu.edu.cn

**Data availability statement:** All relevant data are within the paper and its Supporting

## Abstract

Linguistic Z-numbers (LZs), which uses both fuzzy constraint and reliability measure to describe information and has been widely used in various industries. However, as the amount of information needed to process increases, the relevant sorting methods are inefficient. Three-way decision model with decision-theoretical rough sets divides objects into three disjoint regions, namely acceptance,deferment, and rejection. In the subsequent process, only the objects that need delayed judgment are subdivided, which greatly reduces the amount of computation. Therefore, in order to improve the efficiency of decision making under LZs environment, we propose a sequential three-way decision model. Firstly, by considering both the fuzzy constraint and reliability measure, the three-way decision model with LZs (TWDZ) is proposed. Next, the concept of attribute hierarchy is proposed to prioritize attributes based on contributions to distinguishing alternatives. After that, combined with attribute priority, the objects in delay domain are constantly subdivided, and a sequential three-way decision model is proposed. Finally, considering the application background of double carbon economy, a practical case about selecting an optimal design of electric vehicles charging station is offered, and a comparative analysis was conducted to demonstrate the proposed STWDZ model.

## 1 Introduction

Multi-attribute decision-making (MADM) is a critical research area in modern decision science, where the expression of evaluation values plays a significant role [1,2]. In practical applications, many decision-making activities are challenging to evaluate using precise quantitative methods, leading to the emergence of linguistic term sets (LTSs) [3]. To address the pervasive uncertainty in decision information and enhance evaluation reliability, the theory of linguistic Z-numbers (LZs) [4] was developed. By integrating fuzzy constraints with reliability measures, LZs provide a robust framework for modeling and processing decision information in complex scenarios, making them an essential tool for solving MADM problems [5-8].

Information files. Specifically, the initial data used in this study are presented in Tables 3 and 4, which serve as the foundation for our analysis. The conclusions of the paper can be fully reproduced by applying the proposed model to the data provided in these tables.

**Funding:** This study was supported by the Project of Humanities and Social Sciences Research Project of Chinese Ministry of Education (20XJC790005), Soft Science Research Program of Shaanxi Province (2021KRM091, 2021KRM098). The funders had no role in study design, data collection and analysis, decision to publish, or preparation of the manuscript.

**Competing interests:** The authors have declared that no competing interests exist.

To address the increasing complexity of decision scenarios, various extensions of LZs have been proposed, including hesitant uncertain linguistic Z-numbers (HULZNs) [13], hesitant uncertain discrete linguistic Z-numbers (HUDLZNs) [14], linguistic Z-number fuzzy soft sets (LZFSs) [15]. These extensions are designed to tackle specific challenges by incorporating additional flexibility and refinement into the representation of linguistic information. Their versatility and efficacy have been demonstrated across numerous fields, such as decision-making, risk assessment [9-11] and medical diagnosis [4,12]. By adapting LZs to different decision-making contexts, researchers have enhanced the ability to process linguistic information under conditions of uncertainty and variability.

The application of LZs and their extensions in MADM problems can be broadly categorized into two main approaches. The first involves aggregating evaluation information across all attributes using specialized operators, such as the hesitant uncertain linguistic Z-numbers power-weighted average (HULZPWA) operator, hesitant uncertain linguistic Z-numbers power-weighted geometric (HULZPWG) operator [13], and linguistic Z-number ordered weighted geometric averaging (LZOWGA) operator [11]. The second approach integrates LZs and their variants with established decision-making methods, such as the TODIM method [4] and the VIKOR method [16]. While this integration enhances the applicability and effectiveness of LZs in addressing diverse decision scenarios, it also introduces challenges, such as the involvement of numerous parameters, which increases the complexity and instability of decision outcomes.

In MADM applications, identifying some alternatives as definitively good or bad can eliminate unnecessary evaluations, significantly reducing computational workload. However, traditional methods such as VIKOR [16], which ranks alternatives based on ideal solutions, and PROMETHEE [17, 18], which relies on pairwise dominance comparisons, often require exhaustive evaluations, making them inefficient for large datasets. To address this challenge, researchers have integrated fuzzy information with the three-way decision model (TWDM) [19,20] to solve complex problems across various fields [21-26]. For instance, Liao [27] introduced a TWDM incorporating intuitionistic fuzzy costs based on membership and non-membership degrees. Zhang [28] proposed an intuitionistic fuzzy TWDM considering both optimistic and pessimistic perspectives. Similarly, Dai [29] developed a sequential TWDM using intuitionistic fuzzy and intuitionistic fuzzy distance.

While three-way decision models have been extensively studied, few have integrated linguistic Z-numbers (LZs) into their frameworks. Most existing approaches rely on numerical data or basic linguistic term sets, failing to harness the advantages of LZs, which combine fuzzy constraints with reliability measures. This oversight limits their ability to address the inherent vagueness and uncertainty of real-world decision-making. Hence, this paper combines TWDM to overcome the above deficiencies. The innovativeness of this study can be summarized as follows:

- By considering both the fuzzy constraint and reliability measure, the three-way decision model with LZs (TWDZ) is proposed. TWDZ divides objects into three disjoint regions, namely acceptance,deferment, rejection, which greatly improve the efficiency of decision-making.
- The concept of discrimination index is proposed, after that, attribute weight and attribute hierarchy are proposed to prioritize attributes and remove redundant attributes.
- Based on the TWDZ and attribute hierarchy, a sequential three-way decision model based on linguistic Z-numbers (STWDZ) is established. STWDZ combined TWDZ with attribute priority, the objects in delay domain are constantly subdivided. This can greatly improve the distinction degree between alternatives and reduce computational complexity.

The remainder of the paper is organized as follows: In Sect. 2, some definitions are reviewed briefly. In Sect. 3, the TWDZ model is proposed by considering both the fuzzy constraint and reliability measure of LZs. In Sect. 4, the concept of attribute hierarchy is introduced and STWDZ model is proposed. In Sect. 5, STWDZ model is applied to an illustrative example. In Sect. 6, three comparative analysis are conducted to verify the efficiency and validity of the proposed model. In Sect. 7, some remarkable conclusions are drawn.

## 2 Preliminaries

In this section, some related definitions are briefly reviewed, which are necessary to the subsequent analysis.

### 2.1 Linguistic Z-numbers and its related concepts

**Definition 1** (**Linguistic term set** [2]). *Let $S_{2g} = \{s_i \mid i = 0, 1, 2, \cdots, 2g\}$ be a set with odd cardinality, where $s_i$ represents a possible value for linguistic variables, if the following conditions are satisfied, $S_{2g}$ is called a linguistic term set (LTS).*

$$1.\ \forall i, j \in [0, 2g]\, ; i \leq j \Leftrightarrow s_i \leq s_j$$
$$2.\ \forall i, j \in [0, 2g]\, ; i + j = 2g \Leftrightarrow neg\,(s_i) = s_j$$

*where $neg\,(s_i)$ means the inverse operation of $s_i$.*

In the process of information aggregation, the aggregated results may do not match the elements in the language assessment scale. To preserve all the information provided, the discrete linguistic term set is extended to a continuous one: $\bar{S} = \{s_\alpha | \alpha \in [0, l]\}$, in which $l$ is a sufficiently large positive integer. If $s_\alpha \in S_{2g}$, then it is called an original linguistic term; otherwise, $s_\alpha$ is called a virtual linguistic term.

**Definition 2.** (*linguistic scale function* [23]) *Let $s_i \in S_{2g}$ be a linguistic term, the linguistic scale function (LSF) conducts a mapping f from $s_i$ to $\theta_i$ and the mapping is defined as follows:*

$$f : s_i \rightarrow \theta_i (i = 0, 1, ..., 2g) \tag{1}$$

*where $0 \leq \theta_0 \leq \theta_1 < ... < \theta_{2g} \leq 1$. Clearly, f is absolutely monotonically increasing with the subscript i and illustrates the semantics of $s_i$ in fact.*

**Definition 3** (**linguistic Z-Numbers set** [4]). *Let X be a nonempty finite set, $S_{2t} = \{s_0, s_1, ..., s_{2t}\}$ and $S'_{2l} = \{s'_0, s'_1, ..., s'_{2l}\}$ be two LTSs, which represent different semantic situations. A linguistic Z-numbers set (LZS) on X is defined as a pair of linguistic terms and is written as:*

$$Z = \left\{ \left( x, A_{\phi(x)}, B_{\phi(x)} \right) \middle| x \in X \right\} \tag{2}$$

*where $A_{\phi(x)} \in S_{2g}$ is a fuzzy restriction on the values that the uncertain variable is allowed to take and $B_{\phi(x)} \in S'_{2l}$ is a measure of reliability of the first component. Moreover, $(x, A_{\phi(x)}, B_{\phi(x)})$ can be simplified as $\left( A_{\phi(x)}, B_{\phi(x)} \right)$*

**Definition 4** (**Operations of LZs** [4]). *Let $Z_i = (A_i, B_i)$, $Z_j = \left( A_j, B_j \right)$ be two linguistic Z-numbers, f, g be two LSFs, the some operation are given as:*

$$Z_i \oplus Z_j = \left( f^{-1} \left( f(A_i) + f(A_j) \right), g^{-1} \left( \frac{f(A_i)\, g(B_i) + f(A_j)\, g(B_j)}{f(A_i) + f(A_j)} \right) \right) \tag{3}$$

$$neg(Z_i) = \left( f^{-1} \left( f(A_{2t}) - f(A_i) \right), B_i \right) \tag{4}$$

It is worth noting that the value of $f(A_i) + f(A_j)$ may exceed 1, it becomes a matter of debate which the domain of $f^{-1}$ is $[0,1]$ and it cannot convert values exceed 1 back into language terms. Thus, we define some new operations as follows:

$$Z_i \oplus Z_j = \left( f^{-1} \left( f(A_i) + f(A_j) - f(A_i) f(A_j) \right), g^{-1} \left( g(B_i) g(B_j) \right) \right) \tag{5}$$

the scalar multiplication operation for linguistic Z-numbers is be defined as:

$$kZ_i = \left( f^{-1} \left( 1 - \left( 1 - f(A_i) \right)^k \right), g^{-1} \left( g(B_i)^k \right) \right) \tag{6}$$

In addition, let $Z_i = (A_i, B_i), i = 1, 2, ..., n$ be a collection of n linguistic Z– numbers and $\{w_1, w_2, ..., 2_n\}$ be the corresponding weight, then the linguistic Z-number weighted averaging (LZWA) operator is given as follows:

$$LZWA(Z_1, ..., Z_n) = w_1 Z_1 \oplus w_2 Z_2 \oplus ... \oplus w_n Z_n \tag{7}$$

## 2.2 Some concepts of three-way decisions

The equivalence class and rough membership function are prerequisites of three-way decisions (TWD).

**Definition 5** (**Equivalence relation [19]**). *Given a decision information table $S = (U, AT, V, f)$, where U is a nonempty finite set, AT is an attribute set, $V = \bigcup_{a \in AT} V_a$ is the attribute value set, $V_a$ is a nonempty set of values for attribute a, $f : U \times AT \to V$ represents an information mapping and for any $x \in U, a \in AT$, we have $f(x, a) \in V_a$. Then for any attribute subset $A \subseteq AT$, an equivalence relation $E_A$ is defined as:*

$$xE_A y \Leftrightarrow \left\{ (x, y) \in U \times U \mid \forall a \in A, f(a, x) = f(a, y) \right\}$$

*given an object $x \in U$,*

$$[x] = \{ y \mid y \in U, y E_A x \}$$

*represents the equivalence class under equivalence relation $E_A$.*

**Definition 6** (**Rough membership function [19]**). *Suppose a state set $\{X, X^C\}$ represents that elements are in X and not in X, then the rough membership function of x belonging to X can be calculated by a conditional probability of classification:*

$$Pr(X \mid [x]) = \frac{|X \cap [x]|}{|[x]|} \tag{8}$$

*where $|\cdot|$ denotes the cardinality of a finite set, $[x]$ represents the equivalence class of x. Furthermore the rough membership function of x not belonging to X is calculated by:*

$$Pr(X^C \mid [x]) = 1 - Pr(X \mid [x]) \tag{9}$$

**Definition 7** (**Three-way decision [19]**). *For classifying an object x, suppose a set of actions is $A = \{a_P, a_N, a_B\}$, which corresponds to the acceptance decision $(x \in POS(X))$, deferment*

**Table 1. Cost parameters table.**

|            | $X$            | $X^C$          |
|------------|----------------|----------------|
| $a_P$      | $\lambda_{PP}$ | $\lambda_{PN}$ |
| $a_B$      | $\lambda_{BP}$ | $\lambda_{BN}$ |
| $a_N$      | $\lambda_{NP}$ | $\lambda_{NN}$ |

*decision* $(x \in BND(X))$ *and rejection decision* $(x \in NEG(X))$ *respectively. Moreover, the cost parameters are given by Table Table 1 :*

*where* $\lambda_{PP} < \lambda_{BP} < \lambda_{NP}$ *is the loss function correspond to* $a_P, a_N, a_B$ *when an object belongs to* X; $\lambda_{NN} < \lambda_{BN} < \lambda_{PN}$ *is the loss function correspond to* $a_P, a_N, a_B$ *when an object does not belong to* X. *Furthermore, the decision rules is defined as follows:*

*(1) If* $Pr(X|[x]) \geq \alpha$, *then* $x \in POS(X)$
*(2) If* $\beta < Pr(X|[x]) < \alpha$, *then* $x \in BND(X)$
*(3) If* $Pr(X|[x]) \leq \beta$, *then* $x \in NEG(X)$
*where*

$$\alpha = \frac{\lambda_{PN} - \lambda_{BN}}{(\lambda_{PN} - \lambda_{BN}) + (\lambda_{BP} - \lambda_{PP})}$$

$$\beta = \frac{\lambda_{BN} - \lambda_{NN}}{(\lambda_{BN} - \lambda_{NN}) + (\lambda_{NP} - \lambda_{BP})}$$

# 3 Three-way decision model with linguistic Z-numbers

When dealing with decision problems under LZs environment, it is common to evaluate the relationship between all alternatives, which will be low efficiency when the scale of data increases rapidly. Hence, utilizing the three-way decision theory to divide objects is necessary and will be a more effective approach. In addition, TWD model divides $U$ into three disjoint regions, in the decision-making problem, in addition to efficiency, we also want to a general ordering of the solutions. Therefore, let the above three regions further satisfy some relations are our target.

Suppose the state set is (dominant alternative, non-dominant alternative), then $x_i \in POS(X), x_j \in BND(X), x_k \in NEG(X)$ means to accepting $x_i$ as the dominant alternative, delaying to judge whether $x_j$ is the dominant alternative, and rejecting $x_k$ as the dominant alternative. At this time, $x_i \succ x_j \succ x_k$ is satisfied. Different state sets can also be set according to different situations, in this article, the state set of is assumed to be (dominant alternative, non-dominant alternative).

## 3.1 Model assumption

In this section, some necessary assumptions are proposed.

**Assumption1** Both the attribute values and cost parameter values are linguistic Z-numbers, the Z-cost parameters table is indicated by Table 2.

**Table 2. Z-cost parameters table.**

|         | $X$                        | $X^C$                      |
|---------|----------------------------|----------------------------|
| $a_P$   | $Z_{PP} = (A_{PP}, B_{PP})$ | $Z_{PN} = (A_{PN}, B_{PN})$ |
| $a_B$   | $Z_{BP} = (A_{BP}, B_{BP})$ | $Z_{BN} = (A_{BN}, B_{BN})$ |
| $a_N$   | $Z_{NP} = (A_{NP}, B_{NP})$ | $Z_{NN} = (A_{NN}, B_{NN})$ |

where $A_{\tau\kappa}, B_{\tau\kappa}(\tau = P, B, N, \kappa = P, N)$ are all linguistic terms.

**Assumption 2** First, since 'take false' and 'abandon really' will incur significant costs, therefore,

$$A_{PP} < A_{BP} \ll A_{NP}$$

$$A_{NN} < A_{BN} \ll A_{PN}$$

should be satisfied. In addition, in real life, for a smaller loss, we tend to give greater reliability., so

$$B_{PP} > B_{BP} \gg B_{NP}$$

$$B_{NN} > B_{BN} \gg B_{PN}$$

are also assumed to be true, where $a < b \ll c$ means $b - a < c - b$ and $a > b \gg c$ means $a - b < b - c$.

**Assumption 3** Let $S_{2t}$ and $S'_{2l}$ be two LTSs, $Z_i = (A_i, B_i)$ is a linguistic Z-number, where $A_i \in S_{2t}, B_i \in S'_{2l}$. Then under the benefical criterion, the rough membership of $Z_i$ belonging to $X$ is assumed to be the similarity probability of $Z_i$ and the best evaluation $(s_{2t}, s_{2l}{}')$:

$$E(Z_i) = \frac{f(A_i \wedge s_{2t})}{f(A_i \vee s_{2t})} \cdot \frac{g(B_i \wedge s'_{2l})}{g(B_i \vee s'_{2l})} \tag{10}$$

where $f$ and $g$ are two LSFs, $a \wedge b = min(a, b)$, $a \vee b = max(a, b)$. Under the non-benefical criterion, the rough membership is assumed to be:

$$E(Z_i) = \frac{f(A_{2g-i} \wedge s_{2t})}{f(A_{2g-i} \vee s_{2t})} \cdot \frac{g(B_i \wedge s_{2t})}{g(B_i \vee s'_{2l})} \tag{11}$$

In addition, $1 - E(Z_i)$ can be considered as the rough membership of $Z_i$ not belonging to $X$.

**Assumption 4** Let $R_\tau = (A_\tau, B_\tau), \tau = P, B, N$ be the classification losses, only if $A_P \leq A_B; A_P \leq A_N$ and $B_P \leq B_B; B_P \leq B_N$ are satisfied at the same time, then take $a_P$, namely $x \in POS(X)$, similarly, if $A_N \leq A_B; A_N \leq A_P$ and $B_N \leq B_B; B_N \leq B_P$, then $x \in NEG(X)$.

## 3.2 Model solution

For a linguistic Z-number $(A_{\phi(x)}, B_{\phi(x)})$, denote $E(x)$ is the rough membership degree, then the classification losses are shown as follows:

$$R_P = Z_{PP}E(x) \oplus Z_{PN}(1 - E(x))$$
$$R_B = Z_{BP}E(x) \oplus Z_{BN}(1 - E(x))$$
$$R_N = Z_{NP}E(x) \oplus Z_{NN}(1 - E(x))$$

Based on the novel operations in definition 4

$$R_P = (f^{-1}\left[\left(1 - (1 - f(A_{PP}))^{E(x)}(1 - f(A_{PN}))^{1-E(x)}\right)\right], g^{-1}\left[g(B_{PP})^{E(x)}g(B_{PN})^{1-E(x)}\right])$$
$$R_B = (f^{-1}\left[\left(1 - (1 - f(A_{BP}))^{E(x)}(1 - f(A_{BN}))^{1-E(x)}\right)\right], g^{-1}\left[g(B_{BP})^{E(x)}g(B_{BN})^{1-E(x)}\right])$$
$$R_N = (f^{-1}\left[\left(1 - (1 - f(A_{NP}))^{E(x)}(1 - f(A_{NN}))^{1-E(x)}\right)\right], g^{-1}\left[g(B_{NP})^{E(x)}g(B_{NN})^{1-E(x)}\right])$$

Let $R_\tau = (A_\tau, B_\tau), (\tau = P, B, N)$, combined with assumption **??**, then

$$(1) A_B \leq A_P, A_N \leq A_P; B_B \leq B_P, B_N \leq B_P \Rightarrow x \in POS(X)$$

$$(2) A_P \leq A_B, A_N \leq A_B; B_P \leq B_B, B_N \leq B_B \Rightarrow x \in BND(X)$$

$$(3) A_P \leq A_N, A_B \leq A_N; B_P \leq B_N, B_B \leq B_N \Rightarrow x \in NEG(X)$$

Since $x \in BND(X)$ involves two parameters, first it is discussed. Obviously $f^{-1}$ and $g^{-1}$ are monotonically increasing, denote $f(A_{\tau\kappa}) = a_{\tau\kappa}, g(B_{\tau\kappa}) = b_{\tau\kappa}$, then $A_P \leq A_B, A_N \leq A_B; B_P \leq B_B, B_N \leq B_B$ is equivalent to

$$\left[\left(1 - (1 - a_{BP})^{E(x)}(1 - a_{BN})^{1-E(x)}\right)\right] \leq \left(1 - (1 - a_{pp})^{E(x)}(1 - a_{PN})^{1-E(x)}\right)$$

$$\left[\left(1 - (1 - a_{BP})^{E(x)}(1 - a_{BN})^{1-E(x)}\right)\right] \leq \left(1 - (1 - a_{Np})^{E(x)}(1 - a_{NN})^{1-E(x)}\right)$$

and

$$b_{BP}{}^{E(x)} b_{BN}{}^{1-E(x)} \leq b_{PP}{}^{E(x)} b_{PN}{}^{1-E(x)}; b_{BP}{}^{E(x)} b_{BN}{}^{1-E(x)} \leq b_{NP}{}^{E(x)} b_{NN}{}^{1-E(x)}$$

That is to say

$$\ln\left[(1 - a_{BP})^{E(x)}(1 - a_{BN})^{1-E(x)}\right] \geq \ln\left[(1 - a_{PP})^{E(x)}(1 - a_{PN})^{1-E(x)}\right]$$

$$\ln\left[(1 - a_{BP})^{E(x)}(1 - a_{BN})^{1-E(x)}\right] \geq \ln\left[(1 - a_{NP})^{E(x)}(1 - a_{NN})^{1-E(x)}\right]$$

and

$$\ln\left[b_{BP}{}^{E(x)} b_{BN}{}^{1-E(x)}\right] \leq \ln\left[b_{PP}{}^{E(x)} b_{PN}{}^{1-E(x)}\right];$$

$$\ln\left[b_{BP}{}^{E(x)} b_{BN}{}^{1-E(x)}\right] \leq \ln\left[b_{NP}{}^{E(x)} b_{NN}{}^{1-E(x)}\right]$$

then we will get

$$E(x) \leq \frac{\ln\left(\frac{1-a_{BN}}{1-a_{PN}}\right)}{\ln\left(\frac{1-a_{PP}}{1-a_{BP}} \frac{1-a_{BN}}{1-a_{PN}}\right)}; E(x) \geq \frac{\ln\left(\frac{1-a_{NN}}{1-a_{BN}}\right)}{\ln\left(\frac{1-a_{BP}}{1-a_{NP}} \frac{1-a_{NN}}{1-a_{BN}}\right)}$$

$$E(x) \leq \frac{\ln\left(\frac{b_{BN}}{b_{PN}}\right)}{\ln\left(\frac{b_{PP}}{b_{BP}} \frac{b_{BN}}{b_{PN}}\right)}, E(x) \geq \frac{\ln\left(\frac{b_{NN}}{b_{BN}}\right)}{\ln\left(\frac{b_{BP}}{b_{NP}} \frac{b_{NN}}{b_{BN}}\right)}$$

Let

$$\alpha_1(x) = \frac{\ln\left(\frac{1-a_{BN}}{1-a_{PN}}\right)}{\ln\left(\frac{1-a_{PP}}{1-a_{BP}} \frac{1-a_{BN}}{1-a_{PN}}\right)}, \beta_1(x) = \frac{\ln\left(\frac{1-a_{NN}}{1-a_{BN}}\right)}{\ln\left(\frac{1-a_{BP}}{1-a_{NP}} \frac{1-a_{NN}}{1-a_{BN}}\right)}$$

$$\alpha_2(x) = \frac{\ln\left(\frac{b_{BN}}{b_{PN}}\right)}{\ln\left(\frac{b_{PP}}{b_{BP}} \frac{b_{BN}}{b_{PN}}\right)}, \beta_2(x) = \frac{\ln\left(\frac{b_{NN}}{b_{BN}}\right)}{\ln\left(\frac{b_{BP}}{b_{NP}} \frac{b_{NN}}{b_{BN}}\right)}$$

**Remark 1:** In the same way, if $A_P \leq A_B, B_P \leq B_B, E(x) \geq \alpha_1(x)$ and $E(x) \geq \alpha_2(x)$ will be obtained; if $A_N \leq A_B, B_N \leq B_B, E(x) \leq \beta_1(x)$ and $E(x) \leq \beta_2(x)$ will be obtained. Besides, the relationship between $R_P$ and $R_N$ will be explained in the following theorems.

**Theorem 1:** For the above parameters, the following properties hold.

$$(P1) 0 < \beta_1(x) < \alpha_1(x) < 1; 0 < \beta_2(x) < \alpha_2(x) < 1$$

$$(P2) E(x) \geq \alpha_1(x) \Rightarrow A_P \leq A_B, A_P \leq A_N; E(x) \leq \beta_1(x) \Rightarrow A_N \leq A_B, A_N \leq A_P$$

$$(P3) E(x) \geq \alpha_2(x) \Rightarrow B_P \leq B_B, B_P \leq B_N; E(x) \leq \beta_2(x) \Rightarrow B_N \leq B_B, B_N \leq B_P$$

**Proof:(P1)** Let $f(x) = \frac{\ln x}{\ln(kx)}, f'(x) = \frac{\ln k}{x[\ln(kx)]^2}$, then $f(x)$ is monotonically increasing when $k > 1$. Moreover, according to Assumption 2:

$$0 < a_{PP} < a_{BP} \ll a_{NP} < 1, 0 < a_{NN} < a_{BN} \ll a_{PN} < 1$$

A so

$$1 < \frac{1 - a_{NN}}{1 - a_{BN}} < \frac{1 - a_{BN}}{1 - a_{PN}}; 1 < \frac{1 - a_{PP}}{1 - a_{BP}} < \frac{1 - a_{BP}}{1 - a_{NP}}$$

then

$$0 < \frac{\ln\left(\frac{1-a_{NN}}{1-a_{BN}}\right)}{\ln\left(\frac{1-a_{BP}}{1-a_{NP}}\frac{1-a_{NN}}{1-a_{BN}}\right)} < \frac{\ln\left(\frac{1-a_{BN}}{1-a_{PN}}\right)}{\ln\left(\frac{1-a_{BP}}{1-a_{NP}}\frac{1-a_{BN}}{1-a_{PN}}\right)} \quad < \frac{\ln\left(\frac{1-a_{NN}}{1-a_{BN}}\right)}{\ln\left(\frac{1-a_{PP}}{1-a_{BP}}\frac{1-a_{NN}}{1-a_{BN}}\right)} < 1$$

namely $0 < \beta_1(x) < \alpha_1(x) < 1$. Similarly, according to Assumption 2:A

$$B_{PP} > B_{BP} \gg B_{NP}; B_{NN} > B_{BN} \gg B_{PN}$$

then we will have

$$\frac{b_{BN}}{b_{PN}} > \frac{b_{NN}}{b_{BN}} > 1; 1 < \frac{b_{PP}}{b_{BP}} < \frac{b_{BP}}{b_{NP}}$$

so

$$0 < \frac{\ln\left(\frac{b_{NN}}{b_{BN}}\right)}{\ln\left(\frac{b_{BP}}{b_{NP}}\frac{b_{NN}}{b_{BN}}\right)} < \frac{\ln\left(\frac{b_{BN}}{b_{PN}}\right)}{\ln\left(\frac{b_{BP}}{b_{NP}}\frac{b_{BN}}{b_{PN}}\right)} < \frac{\ln\left(\frac{b_{BN}}{b_{PN}}\right)}{\ln\left(\frac{b_{PP}}{b_{BP}}\frac{b_{BN}}{b_{PN}}\right)} < 1$$

namely $0 < \beta_2(x) < \alpha_2(x) < 1$. Thus, $P1$ have been proved.

**Proof:(P2)**

Obviously $E(x) \geq \alpha_1(x)$, we will get $A_P \leq A_B$, then according to $P1$, $E(x) \geq \alpha_1(x) > \beta_1(x)$, $A_B \leq A_N$ can be obtained. So $A_P \leq A_B$ and $A_P \leq A_N$ are satisfied.

Similarly, $E(x) \leq \beta_1(x)$, we will get $A_N \leq A_B$, then according to $P1$, $E(x) \leq \beta_1(x) < \alpha_1(x)$, $A_B \leq A_P$ can be obtained. So $A_N \leq A_B$ and $A_N \leq A_P$ are satisfied.

**Proof:(P3)**

The proof is the same way as $P2$

**Remark 2** At first, Theorem 1 indicates that the choices $\alpha_i(x)$ and $\beta_i(x)$ (i=1,2) are feasible. In addition, $P2$ and $P3$ actually shows that if $E(x) \geq \alpha_1(x)$ and $E(x) \geq \alpha_2(x)$ are simultaneously satisfied, namely $E(x) \geq max(\alpha_1(x), \alpha_2(x))$, then $A_P \leq A_B, A_P \leq A_N; B_P \leq B_B, B_P \leq B_N$, at this time, $a_P$ should be taken, $x \in POS(X)$. Similarly, if $E(x) \leq min(\beta_1(x), \beta_2(x))$, then $x \in NEG(X)$.

**Remark 3** What can be also obtained from theorem 1 is $min(\beta_1(x), \beta_2(x)) \leq max(\beta_1(x), \beta_2(x)) < min(\alpha_1(x), \alpha_2(x)) \leq max(\alpha_1(x), \alpha_2(x))$. More deeply, the intervals divided according to these four parameters have different properties, which can be vividly shown in Fig 1. where $\sim$ represents that the relationship between the two is indistinguishable. Therefore, for this part of the information, the decision should be deferred, in other words, action $a_B$ should be taken. Thus, if $min(\beta_1(x), \beta_2(x)) \leq E(x) \leq max(\alpha_1(x), \alpha_2(x))$, then $x \in BND(X)$

In combination with Remark 2 and Remark 3, if we denote $\alpha(x) = max(\alpha_1(x), \alpha_2(x))$, $\beta(x) = min(\beta_1(x), \beta_2(x))$ then the definition of three-way decision rules based on linguistic Z-numbers can be given as follows:

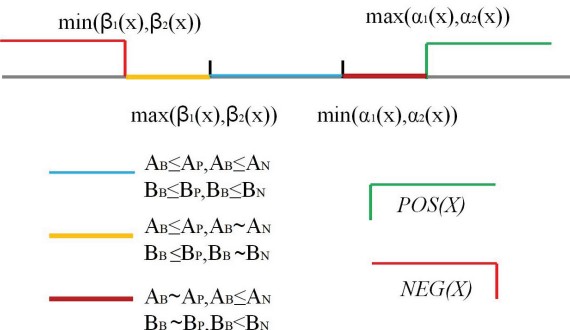

**Fig 1. Interval division caused by different parameter combinations.** Different colored line segments represent the division due to parameters $A_P, A_N, A_B, B_P, B_N, B_B$.

**Definition 8.** *Let* $ZS = (U, AT, LZV, f)$ *be a linguistic Z-numbers information table, where U is a finite nonempty set of objects; AT is a finite nonempty set of attributes, f is an information mapping and LZV is a nonempty set of attribute values which are all represented by LZs. Taking an attribute* $c \in AT$, *an object* $x \in U$, *then the three-way decision model with linguistic Z-numbers (TWDZ) can be defined as:*

*(1) If* $E(x) \geq \alpha(x)$, *then* $x \in POS(X)$

*(2) If* $\beta(x) < E(x) < \alpha(x)$, *then* $x \in BND(X)$

*(3) If* $E(x) \leq \beta(x)$, *then* $x \in NEG(X)$

*where* $E(x)$ *is the rough membership degree of* $f(x,c)$.

## 4 Multi-attribute sequential three-way decision model

Sequential three-way decision (STWD) is a method developed in recent years to deal with uncertain decision. As a concrete model under the concept of granular computing, its goal is to provide a flexible mechanism and method to help users make appropriate decisions in the process of information granulation. STWD consists of a series of TWD whose purpose is to further subdivide the BND by constantly adding new attributes, which has the following form:

Given an information table $(U, AT, V, f)$ and $C_1 \subseteq C_2 \subseteq ... \subseteq C_n \subseteq AT$ be the subset of $AT$. A sequential three-way decision series can be defined as follows

$$STWD = (TWD_1, TWD_2, .., TWD_n)$$

where $TWD_i = (POS_i, BND_i, NEG_i)$, $i = 1, 2, ..., n$ is the three–way decision under attribute set $C_i$, $BND_i$ produced by $TWD_i$ is further divided by $TWD_{i+1}$. The sequential process can be shown vividly in Fig 2.

It can be visually seen that the core idea of $STWD$ is to subdivide $BND_i$. However, it may face such a problem, that is before the algorithm starts, we need to select attribute set: $C_1$, then the subsequent process can be continued, namely the optimal granularity selection problem. Appropriate granularity setting will make the model run more efficiently, on the contrary, if the initial granularity is not reasonable, it will bring great cost to the model. In the sequential process, how to calculate the attribute weights is an essential step. Thus, in this section, the discrimination index is proposed to calculate the attribute weights and a sequential three-way decision model based on linguistic Z-numbers is introduced.

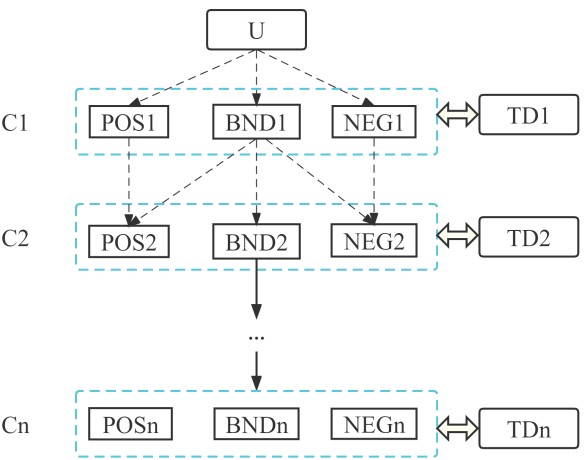

**Fig 2. The process of sequential three-way decisions.** Each step progressively refines the decision-making process, can be described as a top-down approach in which regions are further subdivided at each level.

**Definition 9.** *Let $ZS = (U, AT, LZV, f)$ be a linguistic Z-numbers information table, $ZC = (A_{\tau\kappa}, B_{\tau\kappa})$ $(\tau = P, B, N, \kappa = P, N)$ ba a Z-cost parameters table, where $|AT| = n$, LZVand ZC are all defined on LTS $S_{2t}$ and $S'_{2l}$. Taking an attribute $c_i \in AT, i = 1, 2, ..., n$, an object $x \in U$, $(\alpha, \beta)$ are thresholds calculated from Z-cost parameters table. Then the differentiation index of $c_i$ is defined as:*

$$DI_i = |E(x) > \alpha| + |E(x) < \beta| \tag{12}$$

*where $E(x)$ is the rough membership degree of $f(x, c)$, $|\cdot|$ represent the cardinality of the set.*

*In decision-making problems, a greater contribution to distinguish alternatives should correspond to a greater weight. If the discrimination index of $c_i$ is smaller, it should be assigned a smaller weight, which indicates that the alternatives are analogous for the attribute $c_i$. Otherwise, it should be assigned a larger weight. Thus, assume the attribute weight vector is$\{w_1, w_2, ..., w_n\}$, then $w_i$ can be calculated by:*

$$w_i = \frac{DI_i}{\sum\limits_{i=1}^{n} DI_i} \tag{13}$$

On this basis, the attribute hierarchy can be defined:

**Definition 10.** *Given an attribute set $C = (c_1, c_2, ..., c_n)$ whose weight vector is $\{w_1, w_2, ..., w_n\}$, then the attribute hierarchy can be defined as:*

$$AH_C = (AH_1, AH_2, ..., AH_k)$$

*where $AH_k$ is the set of attribute $c_i \in C$ with the k-th largest weight.*

**Example:** Suppose an attribute set $C = (c_1, c_2, c_3, c_4)$ whose weight vector is $\{0.3, 0.2, 0.25, 0.25\}$, then the attribute hierarchy of $C$ is:

$$AH_C = (\{c_1\}; \{c_3, c_4\}; \{c_2\})$$

The attributes are divided according to the weight from high to low, more specifically, according to the degree of discrimination. On the one hand, attributes that are equally important for distinguishing alternatives are grouped in the same hierarchy, which is actually a granular refinement process. On the other hand, attribute hierarchy with higher discrimination should be given priority to make three-way decisions, which lays a foundation for sequential three-way decisions. In addition, if the discrimination index of $c_i$ is 0, then its weight is also 0 according to Eq. 13, $c_i$ will not appear in the attribute hierarchy, so the attribute hierarchy is also an attribute reduction process.

**Remark 4:** Given $ZS = (U, AT, LZV, f)$ be a linguistic Z-numbers information table, suppose $AH_{AT} = (AH_1, AH_2, ..., AH_k)$ be the attribute hierarchy of $AT$. Taking an object $x \in U$, an attribute hierarchy $AH_k = (c_{k1}, c_{k2}, ..., c_{ks})$. Then these attributes should be normalized first:

$$NF(x, c_{ki})$$
$$= \begin{cases} f(x, c_{ki}), \text{if } c_{ki} \text{ is beneficial} \\ neg(f((x, c_{ki}))), \text{if } c_{ki} \text{ is non-beneficial} \end{cases}$$

and the normalized weight of $c_{ki}$ is

$$\bar{w}_{ki} = \frac{w_{ki}}{\sum_{i=1}^{s} w_{ki}} \tag{14}$$

then we make

$$f(x, AH_k) = LZWA(NF(x, c_{k1}), ..., NF(x, c_{ks})) \tag{15}$$

At this time, the rough membership degree of $AH_k$ can be calculated by Eq 10, and three-way decisions with attribute hierarchy can be converted to TWDZ with single attribute in definition 8.

**Definition 11.** *Let $ZS = (U, AT, LZV, f)$ be a linguistic Z–numbers information table and $AH_{AT} = (AH_1, AH_2, ..., AH_k)$ be the attribute hierarchy of AT, then a sequential three-way decision model with ZS (STWDZ) can be defined as follows*

$$STWDZ = (TWDZ_1, TWDZ_2, ..., TWDZ_k)$$

*where*

$$TWDZ_i = (POS_i, BND_i, NEG_i), i = 1, 2, ..., k$$

*is TWDZ under attribute hierarchy $AH_i$, $BND_i$ produced by $TWDZ_i$ is further divided by $TWDZ_{i+1}$.*

Algorithm 1: Let $ZS = (U, AT, LZV, f)$ be a linguistic Z-numbers information table, where $|U| = n$, LZV are all linguistic Z-numbers defined above $S_{2t}$ and $S'_{2l}$, $AH_{AT} = (AH_1, AH_2, ..., AH_k)$ be the attribute hierarchy of $AT$, $\alpha(x), \beta(x)$ be the two thresholds. Taking an an object $x_i \in U, i = 1, 2, ..., n$, an attribute hierarchy $AH_j \in AH_{AT}, j = 1, 2, ..., k$, if we denote $f(x_i, AH_j) = (A_{ij}, B_{ij})$, then the following algorithm can show the process of a STWDZ model.

It can easily seen that after the above definition, $U$ is divided into three areas regions $POS, BND, NEG$, where

$$POS = \bigcup_{i=1}^{k} POS_i, BND = BND_k, NEG = \bigcup_{i=1}^{k} NEG_i$$

**Algorithm 1. The STWDZ model.**

**Input:** Two thresholds: $\alpha(x), \beta(x)$
Attribute hierarchy: $AH_{AT} = (AH_1, AH_2, ..., AH_k)$
**Output:** POS,BND,NEG

1: initialize $POS = \varnothing, BND = \varnothing, NEG = \varnothing$
2: **for** j from 1 to k **do**
3: $POS_j = \varnothing, BND_j = \varnothing, NEG_j = \varnothing, |U| = n$
4: **for** i from 1 to n **do**
5: $E\left(Z_{ij}\right) = \dfrac{f\left(A_{ij} \wedge s_{2t}\right)}{f\left(A_{ij} \vee s_{2t}\right)} \cdot \dfrac{g\left(B_{ij} \wedge s'_{2l}\right)}{g\left(B_{ij} \vee s'_{2l}\right)}$
6: **if** $E(Z_{ij}) > \alpha(x)$ **then**
7: $POS_j = POS_j \cup \{x_i\}$
8: **else if** $E(Z_{ij}) < \beta(x)$ **then**
9: $NEG_j = NEG_j \cup \{x_i\}$
10: **else**
11: $BND_j = BND_j \cup \{x_i\}$
12: **end if**
13: **end for**
14: $POS = POS \cup POS_j$
15: $NEG = NEG \cup NEG_j$
16: $U = U \cap BND_j$
17: **end for**
18: **return** POS,NEG,BND

in which $POS_i, NEG_i$ is further divided by $BND_{i-1}$, $i = 2, 3, .., k$, we'd like to say compared with $POS_{i+1}$, the elements in $POS_i$ are accepted preferentially. Likewise, the elements in $NEG_i$ are rejected preferentially compared with $NEG_{i+1}$. Thus, following definition is reasonable.

**Definition 12.** *Suppose $x_P^i \in POS_i, x_P^j \in POS_j, x_N^i \in NEG_i, x_N^j \in NEG_j, x_B \in BND, i, j = 1, 2, ..., k$, if $i < j$, then:*

$$x_P^i \succ x_P^j \succ x_B \succ x_N^j \succ x_N^i$$

*are satisfied. Furthermore, if $x_m$ and $x_n$ are divided into the same region under $AH_k$, then*

$$x_m \succ x_n \Leftrightarrow E\left(x_m\right) > E\left(x_n\right)$$

*If $E(x_m) = E(x_n)$ is satisfied at every attribute hierarchy, then we demote $x_m \sim x_n$, where $E(*)$ represents the rough membership degree.*

Summarizing the results above, we come up with a STWDZ model that can be applied to multi-attribute decision making problem under linguistic Z-numbers environment. The decision making procedure can be established as follows:

**Step 1:** Collect the linguistic Z-numbers information table $ZS = (U, AT, LZV, f)$ and Z-cost parameters table $[A_{\tau\kappa}, B_{\tau\kappa}]_{3\times 2}(\tau = P, B, N, \kappa = P, N)$

**Step 2:** Calculate $\alpha(x)$ and $\beta(x)$ on the basis of Z-cost parameters table:

$$\alpha(x) = max\left(\frac{\ln(\frac{1-a_{BN}}{1-a_{PN}})}{\ln\left(\frac{1-a_{PP}}{1-a_{BP}}\frac{1-a_{BN}}{1-a_{PN}}\right)}, \frac{\ln\left(\frac{b_{BN}}{b_{PN}}\right)}{\ln\left(\frac{b_{PP}}{b_{BP}}\frac{b_{BN}}{b_{PN}}\right)}\right)$$

$$\beta(x) = min\left(\frac{\ln\left(\frac{1-a_{NN}}{1-a_{BN}}\right)}{\ln\left(\frac{1-a_{BP}}{1-a_{NP}}\frac{1-a_{NN}}{1-a_{BN}}\right)}, \frac{\ln\left(\frac{b_{NN}}{b_{BN}}\right)}{\ln\left(\frac{b_{BP}}{b_{NP}}\frac{b_{NN}}{b_{BN}}\right)}\right)$$

where $f(A_{\tau\kappa}) = a_{\tau\kappa}, g(B_{\tau\kappa}) = b_{\tau\kappa}, f$ and $g$ are two LSFs.

**Step 3:** Determine the discrimination index of each attribute by Eq 12 and get the attribute hierarchy of $AT$ by definition 10.

**Step 4:** Normalize the attributes at each attribute hierarchy based on remark 4 and aggregate each attribute hierarchy into an overall value by Eq 15.

**Step 5:** Divide $U$ into $POS, BND, NEG$ according to algorithm 1.

**Step 6:** Get the ranking results of alternatives by definition 12.

## 5 An illustrative example

With the continuous progress and development of the current society, the dependence of various industries on energy in the production process has increased, which has further aggravated the energy crisis, and non-renewable energy such as coal and oil is facing the risk of depletion. Therefore, the concept of low-carbon economy has been attached importance to by various countries as soon as it is put forward. In order to achieve the goal of carbon emission reduction, the new energy vehicle market has developed rapidly, because new energy vehicles mainly take electric energy as the main energy supply form, which greatly reduces the impact of vehicle emissions on the environment and is of great significance to environmental protection. As an important component of new energy vehicles, the development speed of electric vehicles is limited by charging facilities and other supporting facilities. Therefore, the design of electric vehicles charging station (EVCS) is particularly important.

To determine an optimal and acceptable EVCS, there are eight alternatives denoted as $U = \{x_1, x_2, .., x_8\}$. And all of alternatives are influenced by five attributes denoted by $AT = \{c_1, c_2, c_3, c_4, c_5\}$, where $c_1$ refers to transaction settlement convenience, $c_2$ refers to charging efficiency, $c_3$ refers to security, $c_4$ refers to charging compatibility, $c_5$ refers to installation and operating costs. The weight vector of five attributes is denoted by $W = (w_1, w_2, w_3, w_4, w_5)$, which are completely unknown. Under two LTS: $S_6$= (very poor, poor, slightly poor, fair, slightly good, good, very good ), $S'_4$=(uncertain, slightly uncertain, medium, slightly sure, sure), we collect the linguistic Z-numbers information table and Z-cost parameters table through a questionnaire survey, which are shown in Tables 3 and 4:

**Step 1:** The two tables are shown in Tables 3 and 4.

**Step 2:** Let $f(s_i) = \frac{i}{6}, i = 1, 2, ..., 6, g(s'_j) = \frac{j}{4}, j = 1, 2, 3, 4$, then we will get :

$$\alpha(x) = 0.7304, \beta(x) = 0.3042$$

**Table 3. Linguistic Z-numbers information table.**

|       | $c_1$        | $c_2$        | $c_3$        | $c_4$        | $c_5$        |
|-------|--------------|--------------|--------------|--------------|--------------|
| $x_1$ | $(s_4, s'_4)$ | $(s_5, s'_2)$ | $(s_6, s'_4)$ | $(s_5, s'_3)$ | $(s_2, s'_2)$ |
| $x_2$ | $(s_6, s'_4)$ | $(s_4, s'_3)$ | $(s_4, s'_2)$ | $(s_5, s'_4)$ | $(s_1, s'_3)$ |
| $x_3$ | $(s_4, s'_3)$ | $(s_3, s'_4)$ | $(s_5, s'_3)$ | $(s_6, s'_3)$ | $(s_2, s'_2)$ |
| $x_4$ | $(s_6, s'_3)$ | $(s_4, s'_4)$ | $(s_4, s'_2)$ | $(s_3, s'_4)$ | $(s_3, s'_4)$ |
| $x_5$ | $(s_5, s'_4)$ | $(s_5, s'_3)$ | $(s_3, s'_4)$ | $(s_4, s'_3)$ | $(s_1, s'_3)$ |
| $x_6$ | $(s_3, s'_3)$ | $(s_3, s'_2)$ | $(s_4, s'_3)$ | $(s_4, s'_2)$ | $(s_3, s'_2)$ |
| $x_7$ | $(s_3, s'_2)$ | $(s_4, s'_2)$ | $(s_5, s'_2)$ | $(s_5, s'_4)$ | $(s_2, s'_2)$ |
| $x_8$ | $(s_5, s'_3)$ | $(s_6, s'_3)$ | $(s_4, s'_4)$ | $(s_5, s'_2)$ | $(s_1, s'_3)$ |

**Table 4. Z-cost parameters table.**

|       | $X$                    | $X^C$                    |
|-------|------------------------|--------------------------|
| $a_P$ | $(s_0, s'_{3.5})$      | $(s_5, s'_1)$            |
| $a_B$ | $(s_2, s'_2)$          | $(s_3, s'_2)$            |
| $a_N$ | $(s_5, s'_1)$          | $(s_{0.5}, s'_{3.5})$    |

**Step 3:** According to Eq. 12 and Eq. 13, The weight vector can be computed:

$$W = (0.36, 0.18.0.09, 0.27, 0.09)$$

and

$$AH_{AT} = (\{c_1\}; \{c_4\}; \{c_2\}; \{c_3, c_5\})$$

can be obtained by definition 10.

**Step 4:** Aggregate each attribute hierarchy into an overall value by Eq 15 and get the attribute hierarchy information table, which are shown in Table 5. ($E(x_i, AH_k)$ denotes rough membership degree).

**Step 5:** Divide $U$ into $POS, BND, NEG$ according to algorithm 1, which can be visualized as Fig 3.

**Step 6:** Get the ranking results of alternatives by definition12.

$$x_2 \succ x_5 \succ x_4 \succ x_3 \succ x_8 \succ x_1 \succ x_6 \succ x_7$$

# 6 Comparative analysis and discussion

In order to further verify the feasibility and validity of the proposed method, we conducted two comparative analysis by applying existing methods to the illustrative example described above.

- Compared with the LZOWGA aggregation operator-based method

In the literature [11], Huang proposed the LZOWGA operator for aggregating linguistic Z-numbers information. In the same example, firstly, the evaluation under the benefit attribute is kept unchanged, and the neg operation of linguistic Z-number is adopted for the evaluation of the non-benefit attribute. Then, the LZOWGA operator is used for aggregating each alternative based on the attribute weights obtained in this paper, the comprehensive

**Table 5. Linguistic Z-numbers information table.**

|       | $AH_1$            | $E(x_i, AH_1)$ | $AH_2$          | $E(x_i, AH_2)$ | $AH_3$          | $E(x_i, AH_3)$ | $AH_4$                      | $E(x_i, AH_4)$ |
|-------|-------------------|----------------|-----------------|----------------|-----------------|----------------|-----------------------------|----------------|
| $x_1$ | $(s_4, s'_4)$     | 0.667          | $(s_5, s'_3)$   | 0.625          | $(s_5, s'_2)$   | 0.427          | $(s_6, s'_{2.828})$         | 0.707          |
| $x_2$ | $(s_6, s'_4)$     | 1.000          | $(s_5, s'_4)$   | 0.833          | $(s_4, s'_3)$   | 0.500          | $(s_{4.586}, s'_{2.449})$   | 0.468          |
| $x_3$ | $(s_4, s'_3)$     | 0.500          | $(s_6, s'_3)$   | 0.750          | $(s_3, s'_4)$   | 0.500          | $(s_{4.586}, s'_{2.449})$   | 0.468          |
| $x_4$ | $(s_6, s'_3)$     | 0.750          | $(s_3, s'_4)$   | 0.500          | $(s_4, s'_4)$   | 0.667          | $(s_{3.551}, s'_{2.828})$   | 0.418          |
| $x_5$ | $(s_5, s'_4)$     | 0.833          | $(s_4, s'_3)$   | 0.500          | $(s_5, s'_3)$   | 0.625          | $(s_{4.268}, s'_{3.464})$   | 0.616          |
| $x_6$ | $(s_3, s'_3)$     | 0.375          | $(s_4, s'_2)$   | 0.333          | $(s_3, s'_2)$   | 0.250          | $(s_{3.551}, s'_{2.449})$   | 0.362          |
| $x_7$ | $(s_3, s'_2)$     | 0.250          | $(s_5, s'_4)$   | 0.833          | $(s_4, s'_2)$   | 0.333          | $(s_{4.586}, s'_2)$         | 0.382          |
| $x_8$ | $(s_5, s'_3)$     | 0.625          | $(s_5, s'_2)$   | 0.417          | $(s_6, s'_3)$   | 0.750          | $(s_{4.586}, s'_{3.464})$   | 0.662          |

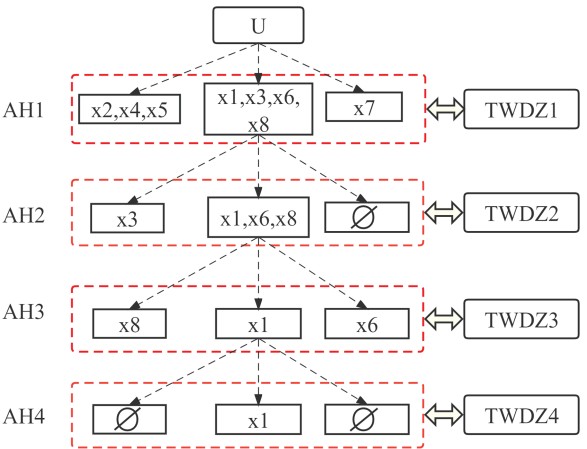

**Fig 3. The process of STWDZ model.** The decision-making process consists of four steps, with each step further processing the options in the intermediate deferred region.

value $r_i$ of alternative $x_i$ will be obtained:

$$r_1 = (s_{4.59}, s'_{3.06}), r_2 = (s_{5.03}, s'_{3.47}), r_3 = (s_{4.33}, s'_{3.05})$$
$$r_4 = (s_{4.18}, s'_{3.38}), r_5 = (s_{4.49}, s'_{3.42}), r_6 = (s_{3.33}, s'_{2.40})$$
$$r_7 = (s_{3.91}, s'_{2.42}), r_5 = (s_{5.06}, s'_{2.76})$$

Moreover, the score of $x_i$ can be calculated:

$$S(x_1) = 0.5859, S(x_2) = 0.7273, S(x_3) = 0.5493,$$
$$S(x_4) = 0.5885, S(x_5) = 0.6398, S(x_6) = 0.3337,$$
$$S(x_7) = 0.3934, S(x_8) = 0.5818$$

Then the ranking results of alternatives can be obtained:

$$x_2 \succ x_5 \succ x_4 \succ x_1 \succ x_8 \succ x_3 \succ x_7 \succ x_6$$

It can be seen that the results obtained by LZOWGA aggregation operator-based method are basically consistent with the ranking results in this paper. However, the aggregation results of alternatives $x_1$, $x_4$ and $x_8$ are less discriminative. With the increase of alternatives and attributes, the the computational complexity of aggregation operator-based method will increase, and the disadvantage of low discrimination will become more obvious. STWDZ model proposed in this paper firstly divides the attributes into a hierarchical level and does not need to calculate all attributes each time the model is used. In addition, for the alternatives in POS and NEG, no further steps are required in the subsequent process, which greatly reduces the complexity of calculation and make the alternatives a greater degree of differentiation.

- Compared with the VIKOR Model.

The VlseKriterijum-ska Optimizacija I Kompromisno Resenje (VIKOR) method, which focuses on choosing options from a set of alternatives. In the context of linguistic Z-numbers, an extended VIKOR model is proposed [13]. Now, the above illustrative example is addressed utilizing Peng's method.

In order to reduce the influence of unnecessary parameters, the attribute weight is assumed to be the weight obtained in this paper, and the distance between $Z_i = (A_i, B_i)$ and $Z_j = (A_j, B_j)$ is defined as: $d(Z_i, Z_j) = |f(A_i)g(B_i) - f(A_j)g(B_j)|$. Then, according to other procedures of the modified VIKOR method in Ref.[13], the closeness coefficients $U(x_i)$ can be computed:

$$U(x_1) = 0.2737, U(x_2) = 0, U(x_3) = 0.4923$$
$$U(x_4) = 0.3147, U(x_5) = 0.2548, U(x_6) = 0.875$$
$$U(x_7) = 0.8764, U(x_8) = 0.3865$$

Then the ranking result can be obtained :

$$x_2 \succ x_5 \succ x_1 \succ x_4 \succ x_8 \succ x_3 \succ x_6 \succ x_7$$

It is mostly consistent with the ranking results in this paper and the best alternative is also $x_2$, which shows the effectiveness of the proposed method. Compared with the aggregation operator-based method, it has a greater discrimination degree for alternatives. However, the ranking method first obtains the group utility and individual regret based on the distance from the positive ideal solution (PIS) and the negative ideal solution (NIS), and then obtains the closeness coefficients. In the decision-making problem, the alternatives often have their own advantages and disadvantages under different attributes, which makes the choice of PIS and the NIS are more extreme. Moreover, in the decision-making step of VIKOR method, many parameters are involved while the variation of parameters in the proposed model only involves the selection of different LSFs. Therefore, STWDZ model is more stability.

With the size of data increases rapidly, the traditional ranking method will have a huge amount of computation, and there may be a problem of low discrimination between the alternatives. In real life, what we need is to choose the better alternatives or discard the worse alternatives. If these alternatives have been selected, there is no need for further calculation. The STWDZ model is based on attribute hierarchy. On the one hand, it is an attribute reduction process. For example, if the discrimination index of $c_i$ is 0, it means that it has no contribution to distinguish alternatives. According to Eq 13, its weight is assigned to 0 and it is not considered in the subsequent calculation, which will greatly reduce the computational complexity. On the other hand, the TWDZ model is only used to further subdivide the alternatives in BND, for the alternatives that have been divided into POS and NEG, there is no need for further calculation, which is more in line with the realistic needs.

- Sensitivity analysis

LSF converts linguistic variables into specific values for calculation. As a bridge between linguistic terms and specific values, it essentially reflects the semantics of the linguistic terms. With different semantic distribution, three different types of LSF are proposed [23]:

*LSF* 1. *f* increases uniformly with the subscript *i*, the assessment scale is divided on average.

$$f_1(s_i) = \theta_i = \frac{i}{2g}, i = 0, 1, ..., 2g$$

*LSF* 2. The growth rate of *f* first decreases and then increases, which indicates that linguistic variables with subscripts away to *g* have a wider semantic coverage, the intermediate linguistic terms are more finely divided.

$$f_2(s_i) = \theta_i$$
$$= \begin{cases} \frac{a^g - a^{g-i}}{2a^g - 2} (i = 0, 1, 2, \cdots, g) \\ \frac{a^g + a^{i-g} - 2}{2a^g - 2} (i = g + 1, g + 2, \cdots, 2g) \end{cases}$$

According to several researches, parameter *a* generally lies in the interval $[1.36, 1.4]$.

*LSF* 3. The growth rate of *f* first increases and then decreases, indicating that linguistic variables with subscripts close to *g* have a wider semantic coverage, the language terms of both ends are more finely divided

$$f_3(s_i) = \theta_i$$
$$= \begin{cases} \frac{g^\alpha - (g-i)^\alpha}{2g^\alpha} (i = 0, 1, 2, \cdots, g) \\ \frac{g^\beta + (i-g)^\beta}{2g^\beta} (i = g + 1, g + 2, \cdots, 2g) \end{cases}$$

Several studies have investigated the parameter and $\alpha = \beta = 0.88$ is determined.

The differences of the three kinds of LSFs are intuitively shown in Fig 4.

To illustrate the influence of the LSFs on the decision-making results, three different LSFs are used to calculate the thresholds $\alpha(x), \beta(x)$ and then rank the alternatives, the ranking results are shown in Table 6 (*a* is set to 1.4 in $f_2(s_i)$).

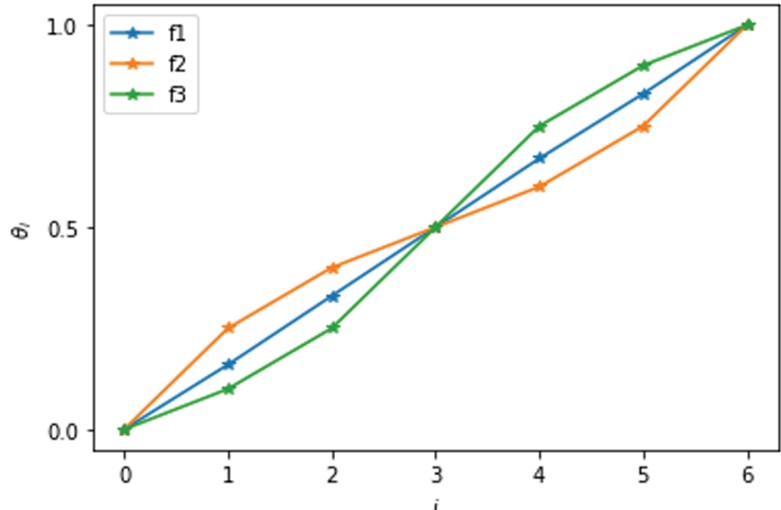

**Fig 4. Illustration of** $f_i(s_i), i = 1, 2, 3f_1(s_i)$ increases uniformly with the subscript *i*; The growth rate of $f_2(s_i)$ first decreases and then increases with the subscript *i*; The growth rate of $f_3(s_i)$ first increases and then decreases.

**Table 6. Ranking results using different LSFs.**

| $f^*(s_i)$ | $g^*(s_i)$ | $\alpha(x)$ | $\beta(x)$ | Ranking results |
|---|---|---|---|---|
| $f_1$ | $f_1$ | 0.7304 | 0.3042 | $x_2 > x_5 > x_4 > x_3 > x_8 > x_1 > x_6 > x_7$ |
| $f_1$ | $f_2$ | 0.7304 | 0.3042 | $x_2 > x_5 > x_4 > x_3 > x_8 > x_1 > x_6 > x_7$ |
| $f_1$ | $f_3$ | 0.7304 | 0.3042 | $x_2 > x_5 > x_4 > x_3 > x_8 > x_1 > x_6 > x_7$ |
| $f_2$ | $f_1$ | 0.6216 | 0.3589 | $x_2 > x_5 > x_8 > x_4 > x_1 > x_3 > x_7 > x_6$ |
| $f_2$ | $f_2$ | 0.6216 | 0.3589 | $x_2 > x_5 > x_8 > x_4 > x_1 > x_3 > x_7 > x_6$ |
| $f_2$ | $f_3$ | 0.6216 | 0.3589 | $x_2 > x_5 > x_8 > x_4 > x_1 > x_3 > x_7 > x_6$ |
| $f_3$ | $f_1$ | 0.7645 | 0.2876 | $x_2 > x_5 > x_8 > x_1 > x_4 > x_3 > x_6 > x_7$ |
| $f_3$ | $f_2$ | 0.7645 | 0.2876 | $x_2 > x_5 > x_8 > x_1 > x_4 > x_3 > x_6 > x_7$ |
| $f_3$ | $f_3$ | 0.7645 | 0.2876 | $x_2 > x_5 > x_8 > x_1 > x_4 > x_3 > x_6 > x_7$ |

From the results in Table 6, we can find that $x_2, x_5$ are always the best two alternatives and $x_6, x_7$ are always the worst two alternatives, the ranking sequences of $\{x_1, x_3, x_4, x_8\}$ are inconsistent with different combinations of LSFs. This is because the thresholds obtained according to Z-cost parameters table will change under different semantic environments. Moreover, under different thresholds, the attribute hierarchy are different and the results of TWDZ model at each hierarchy are also different.

It is worth mentioning that as the semantic terms at both ends become more and more refined, the value of $\alpha(x)$ gradually increases and the value of $\beta(x)$ gradually decreases, which is in line with the meaning of LSF. When the semantic terms at both ends become refined, the requirements of POS and NEG become more stringent, so the thresholds change as described above. In addition, $\alpha(x) = max(\alpha_1(x), \alpha_2(x))$, $\beta(x) = min(\beta_1(x), \beta_2(x))$, in which $\alpha_1(x)$ and $\alpha_2(x)$ are based on LSF $f^*$ and fuzzy restriction $A$, $\beta_1(x)$ and $\beta_2(x)$ are based on LSF $g^*$ and reliability measure $B$, while the result shows that the thresholds are determined by $f^*$, which indicates that there is a potential relationship between fuzzy restriction and reliability measure in linguistics Z-numbers, which is similar to the concept of hidden preference proposed by Zadeh and will be an object of future research in this paper.

In the actual decision-making process, the appropriate LSF can be selected according to different semantic environments. If decision maker is sensitive to the language terms on both sides, LSF like $f_3$ can be taken, if he is sensitive to the language terms in the middle, LSF like $f_2$ can be taken. No matter what LSFs are used, the best alternative is always $x_2$, which verifies the stability and accuracy of the proposed method.

## 7 Conclusion

In the environment of Linguistic Z-numbers, this paper proposes a STWDZ model a based on the idea of minimum loss, which overcomes the shortcomings of traditional methods that have a large amount of calculation and a low degree of discrimination. Firstly, two thresholds of the three-way decision in the Linguistic Z-numbers environment are obtained under reasonable assumptions and TWDZ model are prposed. Next, the concept of attribute hierarchy is proposed based on contributions to distinguishing alternatives, the redundant attributes are

reduced. After that, the TWDZ model is continuously used for alternatives in BND according to the attribute hierarchy. Finally, a practical case about selecting an optimal design of EVCS is offered, and a comparative analysis was conducted to demonstrate the proposed STWDZ model. The results show that the proposed model is practical and flexible, which can not only reduces the computational complexity, but also improves the distinction between alternatives.

In the future, the following aspects are worthy of further study. First, the thresholds obtained are based on the Z-cost parameters table given subjectively, thus, the determination and quantification of these parameters should be considered. Moreover, According to the results, it can be seen that there is a certain connection between $\alpha_1(x)$ and $\alpha_2(x)$. More specifically, there is a connection between the fuzzy limit and the reliability measure of Linguistic Z-number. Therefore, Exploring the potential connection between them is also a future research direction of this paper.

## Acknowledgments

We thank the editorial team and the reviewers for their valuable feedback and constructive suggestions, which have greatly helped to improve the quality of this manuscript.

## Author contributions

**Conceptualization:** Yaning Xu, Yuezhong Fan.

**Methodology:** Yi Mao.

**Resources:** Yi Mao.

**Writing – original draft:** Yi Mao.

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
