## [Decision Letter · Decision Letter 0]

7 Jan 2025

PONE-D-24-37418A sequential three-way decision model based on Linguistic Z-NumbersPLOS ONE

Dear Dr. Mao,

Thank you for submitting your manuscript to PLOS ONE. After careful consideration, we feel that it has merit but does not fully meet PLOS ONE’s publication criteria as it currently stands. Therefore, we invite you to submit a revised version of the manuscript that addresses the points raised during the review process.

Please revise your paper and prepare responses according to the comments from the reviewer and the Editor, which are against the quality of your paper.

We look forward to receiving your revised manuscript.

Kind regards,

Ta-Chung Chu

Academic Editor

PLOS ONE

Journal Requirements:

When submitting your revision, we need you to address these additional requirements. 1. Please ensure that your manuscript meets PLOS ONE's style requirements, including those for file naming. The PLOS ONE style templates can be found at  https://journals.plos.org/plosone/s/file?id=wjVg/PLOSOne_formatting_sample_main_body.pdf and https://journals.plos.org/plosone/s/file?id=ba62/PLOSOne_formatting_sample_title_authors_affiliations.pdf 2. We suggest you thoroughly copyedit your manuscript for language usage, spelling, and grammar. If you do not know anyone who can help you do this, you may wish to consider employing a professional scientific editing service.  The American Journal Experts (AJE) (https://www.aje.com/) is one such service that has extensive experience helping authors meet PLOS guidelines and can provide language editing, translation, manuscript formatting, and figure formatting to ensure your manuscript meets our submission guidelines. Please note that having the manuscript copyedited by AJE or any other editing services does not guarantee selection for peer review or acceptance for publication.  Upon resubmission, please provide the following: • The name of the colleague or the details of the professional service that edited your manuscript• A copy of your manuscript showing your changes by either highlighting them or using track changes (uploaded as a *supporting information* file)• A clean copy of the edited manuscript (uploaded as the new *manuscript* file).  3. We note that your Data Availability Statement is currently as follows:  “All relevant data are within the manuscript and its Supporting Information files.” Please confirm at this time whether or not your submission contains all raw data required to replicate the results of your study. Authors must share the “minimal data set” for their submission. PLOS defines the minimal data set to consist of the data required to replicate all study findings reported in the article, as well as related metadata and methods (https://journals.plos.org/plosone/s/data-availability#loc-minimal-data-set-definition). For example, authors should submit the following data: - The values behind the means, standard deviations and other measures reported;- The values used to build graphs;- The points extracted from images for analysis. Authors do not need to submit their entire data set if only a portion of the data was used in the reported study. If your submission does not contain these data, please either upload them as Supporting Information files or deposit them to a stable, public repository and provide us with the relevant URLs, DOIs, or accession numbers. For a list of recommended repositories, please see https://journals.plos.org/plosone/s/recommended-repositories. If there are ethical or legal restrictions on sharing a de-identified data set, please explain them in detail (e.g., data contain potentially sensitive information, data are owned by a third-party organization, etc.) and who has imposed them (e.g., an ethics committee). Please also provide contact information for a data access committee, ethics committee, or other institutional body to which data requests may be sent. If data are owned by a third party, please indicate how others may request data access. 4. PLOS requires an ORCID iD for the corresponding author in Editorial Manager on papers submitted after December 6th, 2016. Please ensure that you have an ORCID iD and that it is validated in Editorial Manager. To do this, go to ‘Update my Information’ (in the upper left-hand corner of the main menu), and click on the Fetch/Validate link next to the ORCID field. This will take you to the ORCID site and allow you to create a new iD or authenticate a pre-existing iD in Editorial Manager. 5. Please update your submission to use the PLOS LaTeX template. The template and more information on our requirements for LaTeX submissions can be found at http://journals.plos.org/plosone/s/latex. 6. We note you have included a table to which you do not refer in the text of your manuscript. Please ensure that you refer to Table 2 in your text; if accepted, production will need this reference to link the reader to the Table.

**Additional Editor Comments:**

Please consider the following suggestions to revise your work for possible publication.

The introduction is too long. Introduction and literature review should be separated into two independent sections.The introduction is to precisely present the research’s background, purpose, the advantages of the proposed method compared to the existing papers, findings of comparisons, and paper structure, etc.A more in-depth literature review of relevant methods, including LZs, HULZNs, HUDLZNs, LZFSs, PROMETHEE, PROMETHEEII, HULZPWA, HULZPWG, LZOWGA, TODIM, VIKOR, TWDM, etc., should be conducted to clearly show the research gap that needs to be addressed; for example, TWDM related methods should be briefly introduced. In section 6, a comparison with one method, i.e., LZOWGA aggregation operator-based method, is insufficient. A comparison with more methods needs to be made to display the effectiveness of the proposed method. Similarly, more methods, other than only VIKOR, need to compared.The most recently cited paper was published in 2022, which is old and should be updated to 2024.

Reviewers' comments:

Reviewer's Responses to Questions

**Comments to the Author**

1. Is the manuscript technically sound, and do the data support the conclusions?

Reviewer #1: Yes

2. Has the statistical analysis been performed appropriately and rigorously? 

Reviewer #1: Yes

3. Have the authors made all data underlying the findings in their manuscript fully available?

Reviewer #1: Yes

4. Is the manuscript presented in an intelligible fashion and written in standard English?

Reviewer #1: Yes

5. Review Comments to the Author

Reviewer #1: Overall, paper is well written. The method is well explained. However, the motivation on why the suggested method necessary is vague, not explained.

Page 13: Bring Step 1 to next paragraph

Fig.2 & Fig.3 exactly the same figure. If they are the similar figure, just show once and recall the figure for other sections.

6. PLOS authors have the option to publish the peer review history of their article (what does this mean?). If published, this will include your full peer review and any attached files.

Reviewer #1: No

---

## [Author Response · Author response to Decision Letter 1]

20 Jan 2025

Thank you very much for your valuable feedback and for taking the time to review our manuscript. We greatly appreciate the thoughtful comments and suggestions provided, which have significantly contributed to improving the quality of our work.

We have carefully considered each point raised by the reviewers and the editor and have made revisions accordingly to address the concerns and strengthen the manuscript. Modifications made in response to the editor’s suggestions are highlighted in blue, while those addressing the reviewers’ comments are highlighted in red. Details of the changes can be found in the marked-up version of the revised manuscript.

---

## [Decision Letter · Decision Letter 1]

18 Feb 2025

A sequential three-way decision model based on Linguistic Z-numbers

PONE-D-24-37418R1

Dear Dr. Mao,

We’re pleased to inform you that your manuscript has been judged scientifically suitable for publication and will be formally accepted for publication once it meets all outstanding technical requirements.

Kind regards,

Ta-Chung Chu

Academic Editor

PLOS ONE

Additional Editor Comments (optional):

Reviewers' comments:

Reviewer's Responses to Questions

**Comments to the Author**

1. If the authors have adequately addressed your comments raised in a previous round of review and you feel that this manuscript is now acceptable for publication, you may indicate that here to bypass the “Comments to the Author” section, enter your conflict of interest statement in the “Confidential to Editor” section, and submit your "Accept" recommendation.

Reviewer #1: All comments have been addressed

2. Is the manuscript technically sound, and do the data support the conclusions?

Reviewer #1: Yes

3. Has the statistical analysis been performed appropriately and rigorously? 

Reviewer #1: Yes

4. Have the authors made all data underlying the findings in their manuscript fully available?

Reviewer #1: Yes

5. Is the manuscript presented in an intelligible fashion and written in standard English?

Reviewer #1: Yes

6. Review Comments to the Author

Reviewer #1: (No Response)

7. PLOS authors have the option to publish the peer review history of their article (what does this mean?). If published, this will include your full peer review and any attached files.

Reviewer #1: No

---

## [Editor Report · Acceptance letter]

PONE-D-24-37418R1

PLOS ONE

Dear Dr. Mao,

I'm pleased to inform you that your manuscript has been deemed suitable for publication in PLOS ONE. Congratulations! Your manuscript is now being handed over to our production team.

Kind regards,

on behalf of

Dr. Ta-Chung Chu

Academic Editor

PLOS ONE